# Absent in Melanoma 2 (AIM2) Regulates the Stability of Regulatory T Cells

**DOI:** 10.3390/ijms23042230

**Published:** 2022-02-17

**Authors:** Beatriz Lozano-Ruiz, Amalia Tzoumpa, Claudia Martínez-Cardona, David Moreno, Ana M. Aransay, Ana R. Cortazar, Joanna Picó, Gloria Peiró, Juanjo Lozano, Pedro Zapater, Rubén Francés, José M. González-Navajas

**Affiliations:** 1Alicante Institute for Health and Biomedical Research (ISABIAL), Hospital General Universitario Dr. Balmis, 03010 Alicante, Spain; lozano_bea@gva.es (B.L.-R.); amaliatzoumpa21@gmail.com (A.T.); cmc.cardona@gmail.com (C.M.-C.); david.moreno05@alu.umh.es (D.M.); joanna_pc@hotmail.com (J.P.); peiro_glo@gva.es (G.P.); zapater_ped@gva.es (P.Z.); rfrances@umh.es (R.F.); 2Networked Biomedical Research Center for Hepatic and Digestive Diseases (CIBERehd), Institute of Health Carlos III, 28029 Madrid, Spain; amaransay@cicbiogune.es (A.M.A.); juanjo.lozano@ciberehd.org (J.L.); 3Center for Cooperative Research in Biosciences (CIC bioGUNE), 48160 Derio, Spain; acortazar@cicbiogune.es; 4Pathology Unit, Hospital General Universitario Dr. Balmis, 03010 Alicante, Spain; 5Department of Pharmacology, Pediatrics and Organic Chemistry, University Miguel Hernández (UMH), 03202 Elche, Spain; 6Institute of Research, Development and Innovation in Healthcare Biotechnology in Elche (IDiBE), University Miguel Hernández (UMH), 03202 Elche, Spain; 7Department of Clinical Medicine, University Miguel Hernández (UMH), 03202 Elche, Spain

**Keywords:** absent in melanoma 2, AIM2, inflammasome, T cell, CD4^+^, regulatory T cell, Treg, FOXP3

## Abstract

Absent in melanoma 2 (AIM2) is a cytosolic dsDNA sensor that has been broadly studied for its role in inflammasome assembly. However, little is known about the function of AIM2 in adaptive immune cells. The purpose of this study was to investigate whether AIM2 has a cell-intrinsic role in CD4^+^ T cell differentiation or function. We found that AIM2 is expressed in both human and mouse CD4^+^ T cells and that its expression is affected by T cell receptor (TCR) activation. Naïve CD4^+^ T cells from AIM2-deficient (*Aim2^−/−^*) mice showed higher ability to maintain forkhead box P3 (FOXP3) expression in vitro, while their capacity to differentiate into T helper (Th)1, Th2 or Th17 cells remained unaltered. Transcriptional profiling by RNA sequencing showed that AIM2 might affect regulatory T cell (Treg) stability not by controlling the expression of Treg signature genes, but through the regulation of the cell’s metabolism. In addition, in a T cell transfer model of colitis, *Aim2^−/−^*-naïve T cells induced less severe body weight loss and displayed a higher ability to differentiate into FOXP3^+^ cells in vivo. In conclusion, we show that AIM2 function is not confined to innate immune cells but is also important in CD4^+^ T cells. Our data identify AIM2 as a regulator of FOXP3^+^ Treg cell differentiation and as a potential intervention target for restoring T cell homeostasis.

## 1. Introduction

Absent in melanoma 2 (AIM2) is a cytosolic receptor that senses dsDNA from any origin, including viruses, bacteria and self-dsDNA [1,2,3]. Although it was originally identified as a tumor suppressor in melanoma [4], most of the research conducted on AIM2 has focused on its function in inflammasome activation and innate immune defense against intracellular pathogens [5]. The inflammasome is a cytosolic protein complex that promotes the recruitment and activation of caspase-1, the subsequent release of inflammatory cytokines such as interleukin (IL)-1β and IL-18, and a form of cell death known as pyroptosis [6,7]. In addition to AIM2, various pattern recognition receptors can induce the assembly of the inflammasome complex, including several members of the NOD-like receptor (NLR) family, the AIM2-like receptor known as interferon-γ (IFNγ)-inducible protein 16 (IFI16), and the protein Pyrin [8,9]. Each of these receptors is activated by different stimuli, from microbial components to signs of cellular disturbance or damage. Among all these receptors, the NLRP3 (NLR family, pyrin domain containing 3) is the prototypical and most thoroughly studied inflammasome. Once they are activated, the majority of these receptors, including AIM2 and NLRP3, require the adaptor protein ASC (apoptosis-associated speck-like protein, containing a caspase recruitment domain) to recruit caspase-1 and initiate full inflammasome assembly [10]. Since its initial description in 2002 [6], thousands of studies have uncovered the role of inflammasomes in innate immunity, in multiple inflammatory and autoimmune diseases, and in many tumors. However, very few studies have investigated the direct role of inflammasome-associated receptors in adaptive immune cells. Nonetheless, emerging evidence is highlighting the importance of these receptors in CD4^+^ T helper (Th) cell differentiation and function. CD4^+^ T cells from ASC-deficient mice were shown to produce reduced amounts of IFNγ and elevated amounts of IL-10 when stimulated in vitro [11]. The adaptor ASC has also been reported to play a critical T-cell-intrinsic role in Th17-mediated experimental autoimmune encephalomyelitis, which was attributed to IL-1β production by polarized Th17 cells via ASC-NLRP3 inflammasome [12]. In another study, NLRP3 was shown to control mouse Th2 differentiation in a cell-intrinsic but inflammasome-independent manner [13]. There is also evidence of NLRP3 involvement in Th1 differentiation of human CD4^+^ T cells [14]. In summary, these reports highlight the importance of ASC and NLRP3 in the differentiation or function of Th1, Th2 and Th17 cells.

In addition to effector Th cells, CD4^+^ cells can differentiate into regulatory T (Treg) cells. Treg cells are essential regulators of immune homeostasis, characterized by a distinct pattern of gene expression and epigenetic modifications that includes upregulation of immune suppressive genes and downregulation of inflammatory genes [15]. Epigenetic changes, however, are not specific attributes of Treg cells, since they greatly influence the function and differentiation of all CD4^+^ T cell subsets [16,17,18]. Although Treg cells are not an homogenous population, the transcription factor forkhead box P3 (FOXP3) is considered as the master regulator of Treg cell differentiation, maintenance and function [19,20]. FOXP3 expression is induced in some CD4^+^ T cells during their thymic development, and these cells have been termed thymus-derived Tregs (tTreg). FOXP3^+^ cells may also arise from naïve CD4^+^ T cells at the periphery after antigen stimulation in the presence of an appropriate combination of cytokines, including IL-2 and transforming growth factor (TGF)-β. These cells have been termed induced Treg (iTreg) cells when generated in vitro and peripherally induced Treg (pTreg) cells when generated in vivo [20]. In addition, Treg cells display tissue-specific adaptation and heterogeneity, showing different gene expression patterns and effector molecules at different tissue localization points [21]. 

Although Treg cells have been the subject of intense research efforts, the mechanisms that govern Treg differentiation and function still require further investigation. Recently, an inflammasome-independent function of AIM2 involved in promoting Treg cell stability has been reported [22]. Here, we report that AIM2 is expressed in both human and mouse CD4^+^ T cells and that its expression is regulated by T cell receptor (TCR) activation. We also show that lack of AIM2 also supports the stability FOXP3^+^ cells in vitro and their differentiation in vivo during a naïve T cell transfer model of colitis. These data suggest that AIM2 may act as a negative regulator of Treg cells.

## 2. Results

### 2.1. AIM2 Expression in CD4^+^ T Cells Is Controlled by TCR Activation

We first assessed AIM2 expression in both human and mouse CD4^+^ T cells. AIM2 mRNA and protein expression was detected in CD3^+^CD4^+^ cells from the peripheral blood of healthy subjects by qRT-PCR amplification (Figure 1a), flow cytometry (Figure 1b) and immunoblot (Figure 1c). To further investigate the expression and regulation of AIM2 in highly pure mouse CD4^+^ cells, we FACS-sorted CD3^+^CD4^+^TCRβ^+^ cells from the spleens of C57BL/6 mice (purity > 98.5% in all experiments) (Appendix A). At the mRNA level, AIM2 mRNA was detected in all samples after qPCR amplification (Figure 1d). At the protein level, AIM2 protein expression was transiently increased after TCR activation with anti-CD3 and anti-CD28 antibodies (Figure 1e). Since the activation of toll-like receptor 4 (TLR4) has been shown to affect CD4^+^ T cell activation and TCR signaling [23], we investigated whether TLR4 activation also affects the TCR-dependent AIM2 expression. However, a similar AIM2 expression pattern was obtained in CD4^+^ cells pre-stimulated with lipopolysaccharide (LPS) and then treated with anti-CD3/CD28 antibodies (Figure 1e), suggesting that TLR4 does not have a major effect on AIM2 expression in these cells. In summary, these data indicate that AIM2 is expressed in both human and mouse CD4^+^ T cells and that its expression, at least in mouse T cells, is affected by TCR signaling.

### 2.2. AIM2 Deficiency Facilitates the Stability of iTreg Cells

We next aimed to investigate whether the lack of AIM2 affects CD4^+^ T cell functions such as proliferation, differentiation and cytokine production. No significant changes in proliferation were detected after in vitro stimulation of CFSE-labeled splenic CD4^+^ cells for 48 or 72 h (Appendix A). *Aim2^−/−^* CD4^+^ T cells also showed comparable activation-induced cell death, as evidenced by the similar percentages of late-stage apoptotic cells (Annexin V^+^ and 7-AAD^+^) and total Annexin V^+^ cells after 24 h of TCR stimulation with anti-CD3/CD28 antibodies (Appendix A). We then evaluated the influence of AIM2 on the production of pro- and anti-inflammatory cytokines after TCR stimulation. Of note, CD4^+^ T cells from *Aim2^−/−^* mice secreted normal levels of Th1, Th2 and Th17 key cytokines, such as IFNγ, IL-4 and IL-17A, respectively (Figure 2a). By contrast, these cells produced elevated amounts of IL-10 when compared with CD4^+^ T cells from wild-type (WT) mice (Figure 2a). In addition, TCR activation with lower concentrations (0.5, 1 and 2 μg/mL) of anti-CD3 antibody (in the presence of a fixed concentration of soluble anti-CD28) also showed increased IL-10 production by *Aim2^−/−^* T cells, while no differences were detected in IL-2 or IFNγ production (Appendix A). Since IL-10 plays an important role in the immune-suppressive functions of Treg cells, we asked whether AIM2 might be involved in Treg cell differentiation. We first analyzed the mRNA expression of the main transcription factors involved in CD4^+^ T cell differentiation. Notably, expression of *Foxp3*, a key transcription factor for Treg differentiation and function, was upregulated in *Aim2^−/−^* T cells after 24 h of TCR stimulation (Figure 2b). Moreover, the canonical Th2 transcription factor GATA binding protein 3 (*Gata3*), which also supports Treg function [24,25], was also upregulated in *Aim2^−/−^* T cells. By contrast, the expression of *Tbx21* (T-bet) and *Rorc* (RORγt) mRNA, key transcription factors of Th1 and Th17 cells, remained unaltered (Figure 2b).

To further delineate whether AIM2 is involved in T cell differentiation, particularly in Treg cells, we isolated naïve CD4^+^ T cells (CD4^+^CD44^low^CD62L^high^) from the spleens of WT and *Aim2^−/−^* mice and cultured them under Th1, Th2 Th17 or Treg differentiation conditions. Consistent with the profile of cytokine production by total CD4^+^ cells (Figure 2a), we did not find significant differences in the generation of IFNγ^+^, IL-4^+^ or IL-17A^+^ cells (Appendix A), indicating that AIM2 does not participate in the differentiation of these Th cell subsets in vitro. When cultured under Treg differentiation conditions, no differences were observed between WT and *Aim2^−/−^* T cells at day 3 of culture (Figure 2c,d), since almost all cells stained positive for FOXP3. However, *Aim2^−/−^* T cells retained higher expression levels of FOXP3 after 5 days of culture in these conditions (Figure 2c,d). We next examined whether AIM2 also affects the suppressive capacity of these cells. However, an in vitro suppression assay showed that *Aim2^−/−^* and WT Treg cells were equally effective at reducing the proliferation of naïve T cells at all ratios tested (Figure 2e,f), suggesting that AIM2 does not modify the suppressive ability of Treg cells in vitro. Last, higher levels of AIM2 mRNA expression have been detected in freshly isolated Treg cells in comparison with bulk CD4^+^ T cells [22], but protein expression in these and other specific Th subsets has not been investigated. We, therefore, analyzed AIM2 protein expression by flow cytometry in Th1, Th2, Th17 and Treg cells generated in vitro. Intriguingly, similar expression levels were detected in all culture conditions (Appendix A), suggesting that AIM2 expression is similarly retained in Th and Treg cells. Collectively, these data suggest that AIM2 might affect the ability to maintain FOXP3 expression and regulate Treg stability without modifying their suppressive capacity.

### 2.3. RNAseq and GSEA Analysis Reveals Different Metabolic Gene Signatures in Aim2^−/−^ T Cells

To identify possible mechanisms whereby the absence of AIM2 might favor FOXP3 stability, we analyzed the cellular transcriptome from naïve CD4^+^ T cells and from CD4^+^ T cells undergoing Treg differentiation in vitro (2 days after stimulation and culture in Treg conditions) by RNA sequencing (RNAseq) and gene set enrichment analysis (GSEA). Among the most enriched gene sets, we found that AIM2 deficiency led to enhanced gene signatures associated with oxidative phosphorylation and mitochondrial respiration (Figure 3a), both of which are known to promote Treg differentiation and function [26,27,28,29,30]. Notably, these gene sets were enriched in both *Aim2^−/−^* naïve T cells and *Aim2^−/−^* T cells stimulated under Treg-polarizing conditions (Figure 3a). On the other hand, the gene signature associated with glycolysis, which negatively affects Treg cell function [26,29,31], was significantly downregulated in *Aim2^−/−^* naïve T cells (Figure 3b). Additionally, lack of AIM2 during Treg differentiation led to the enrichment of a gene set associated with the activity of phosphatase and tensin homolog (PTEN) (Figure 3c), which is also important for maintaining Treg function and stability [32,33].

Although Treg cells are phenotypically and functionally heterogenous, a small Treg signature gene set has been identified by single-cell RNAseq (sc-RNAseq) [21,34]. This Treg signature includes genes such as *Il2ra*, *Il2rb*, *Ctla4*, *Ikzf2*, *Nrp1*, *Capg*, *Pdcd1*, *Ebi3*, *Itgae*, *Klrg1*, *Gzmb*, and *Penk*, as well as members of the TNFR superfamily such as *Tnfrsf4* and *Tnfrsf9* [21,34]. Separate analysis of this gene set displayed no obvious differences between naïve T cells from WT and *Aim2^−/−^* mice, as shown in the heatmap (Figure 3d), although a small increase in the expression of these genes was suggested by the GSEA analysis (Figure 3e). More importantly, no difference was observed in T cells stimulated under Treg conditions (Figure 3d,e), indicating that AIM2 deficiency does not significantly affect the expression of this Treg-associated gene signature. Together, these data suggest that AIM2 might affect Treg stability not by controlling the expression of Treg signature genes, but through the regulation of the T cell metabolic program.

### 2.4. Aim2^−/−^ Naïve T Cells Show Enhanced Differentiation into FOXP3^+^ Cells after Adoptive Transfer

To determine whether AIM2 could also affect the differentiation or inflammatory potential of naïve T cells in vivo, we employed a well-known adoptive transfer model of colitis [35,36]. We transferred naïve CD4^+^ T cells (CD4^+^CD45RB^high^CD25^−^) from WT or *Aim2^−/−^* donor mice to *Rag1^−/−^* recipients, which lack mature T and B cells, and evaluated the abundance of Treg cells and colitis development after a follow-up period of 15 weeks. Moreover, to evaluate possible differences in the immunosuppressive function of *Aim2^−/−^* Treg cells in vivo, we also cotransferred Treg cells (CD4^+^CD45RB^low^CD25^+^) from WT or *Aim2^−/−^* mice along with the naïve CD4^+^ subset from WT mice. Both types of Treg cells, either from WT or *Aim2^−/−^* mice, were equally efficient at inhibiting the development of colitis in the *Rag1^−/−^* recipients, as shown by body weight data (Figure 4a) and gross examination of the colons (Appendix A). However, when naïve CD4^+^ cells were transferred alone, naïve cells from *Aim2^−/−^* mice induced less body weight loss than those from WT mice (Figure 4b). In addition, gross examination of the colons revealed less severe shortening and bowel wall thickening in *Rag1^-/-^* mice transferred with *Aim2^−/−^* naïve T cells (Figure 4c and Appendix A). Histopathological analysis of the colons also revealed a moderate reduction in the colitis severity score in mice receiving *Aim2^−/−^* naïve T cells when compared with mice receiving WT naïve T cells, although the difference did not reach the statistical significance threshold (Figure 4d) (*p =* 0.07). We could identify areas of severe intestinal inflammation in 6 out of 14 *Rag1^−/−^* mice receiving *Aim2^−/−^* naïve T cells (42.8%), and in 8 out of 14 *Rag1^−/−^* mice receiving WT naïve T cells (57.1%) (Figure 4e). These areas were characterized by very severe inflammatory infiltrate, epithelial cell hyperplasia, crypt abscesses, transmural infiltration and severe loss of crypt structure. Consistent with the body weight data, very few or no signs of intestinal inflammation were observed in *Rag1^−/−^* mice cotransferred with either WT or *Aim2^−/−^* Treg cells (Figure 4d,e).

To examine whether naïve CD4^+^ T cells from *Aim2^−/−^* mice also show a higher tendency towards Treg differentiation in vivo, we isolated CD4^+^ cells from the spleens of the recipient mice and measured FOXP3 expression by flow cytometry. Importantly, *Rag1^−/−^* mice transferred with *Aim2^−/−^* naïve CD4^+^ T cells showed a higher number of CD4^+^FOXP3^+^ cells in spleen compared with mice transferred with WT naïve T cells (Figure 5a,b), indicating that the absence of AIM2 facilitates the stable conversion of naïve CD4^+^ T cells into Treg cells after adoptive transfer. No significant difference was observed between the cotransfer groups (Figure 5a). In agreement with these data, total CD4^+^ T cells isolated from the spleens of mice transferred with *Aim2^−/−^* naïve CD4^+^ T cells also showed significantly higher IL-10 production after in vitro stimulation with anti-CD3 and anti-CD28 antibodies (Figure 5c). Moreover, some of these cells also produced high amounts of IFNγ and slightly elevated amounts of IL-17A upon stimulation, but these differences were not consistent or statistically significant (Figure 5d). Next, we evaluated the presence of FOXP3^+^ cells in the colons of the recipient mice. Importantly, we observed that a higher number of FOXP3^+^ cells were present in the inflamed areas of the colon of mice receiving *Aim2^−/−^* naïve T cells compared with mice receiving WT naïve T cells (Figure 5e,f), although the difference did not reach the statistical significance threshold (*p =* 0.053). Collectively, these data indicate that the lack of AIM2 facilitates the appearance of CD4^+^FOXP3^+^ cells from naïve CD4^+^ T cells in vivo and ameliorates body weight loss in adoptively transferred *Rag1**^−/−^* mice, but is not sufficient to fully prevent intestinal inflammation in this model. 

## 3. Discussion

Over the last decade, numerous studies have documented the importance of AIM2 in inflammation, immune defense against intracellular pathogens and cancer. Most of that research was focused on either innate immune cells or non-immune cells. However, very few studies have addressed the expression and cell-intrinsic function of inflammasome-associated receptors in CD4^+^ T cells. Here, we provide evidence of the role of AIM2 in CD4^+^ T cell differentiation and function. We show that AIM2 is expressed in CD4^+^ T cells and that such expression varies depending on TCR activation. Our data also indicate that AIM2 might be important in iTreg and pTreg biology. In both naïve CD4^+^ T cells and during Treg differentiation, AIM2 deficiency results in a gene expression program that favors oxidative phosphorylation and mitochondrial respiration (Appendix A), which are metabolic traits that enhance Treg stability and function. In addition, lack of AIM2 also promotes the stability of FOXP3^+^ cells in vitro and their differentiation in vivo in a T cell transfer model. In summary, these data suggest that AIM2 could be negatively regulating FOXP3 expression and Treg cell stability. However, the mechanistic details explaining how AIM2 carries out these functions are still unknown.

Previously, ASC deficiency resulted in higher IL-10 production from both total CD4^+^ cells and CD44^high^CD25^+^ cells, although FOXP3 expression remained unchanged in these cells with or without TCR stimulation [11]. These data suggest that ASC may affect cytokine production by already established Treg cells or by FOXP3^−^ T cells, but it also suggests that ASC does not control FOXP3 expression or FOXP3^+^ Treg differentiation. Therefore, since AIM2 strictly requires ASC to recruit caspases and initiate inflammasome assembly, these data already suggest that the effect of AIM2 on FOXP3 expression might be inflammasome-independent. In fact, very recently, Chou et al. also showed that AIM2 has an intrinsic and inflammasome-independent role in Treg cells [22]. In this study, however, AIM2 promoted Treg stability and function by increasing lipid oxidation and attenuating the AKT-mTOR pathway and glycolysis [22]. Although we do not have a clear explanation for these apparently contradictory results, they might be due to the use of genetically different AIM2-deficient mice in both studies. In addition, other methodological differences also exist that may partly explain the divergent results in some experiments, such as the use of a different T cell population as a starting point for in vitro Treg cell subset differentiation and for RNAseq analysis, since we used naïve CD4^+^ T cells whereas Chou et al. used total CD4^+^ cells for these experiments. Higher AIM2 mRNA expression in Treg cells compared to conventional T cells was also reported by Chou et al. [22], while we did not find differences in AIM2 protein expression among Treg, Th1, Th2, and Th17 cells generated in vitro. This disparity may be due to the analysis of different molecules (mRNA vs. protein) or the use of Treg cells from different origins (in-vivo- vs. in-vitro-generated).

In our experiments using the naïve T cell transfer model, *Aim2^−/−^* naïve CD4^+^ T cells differentiated more efficiently into FOXP3^+^ cells in vivo and produced large amounts of IL-10, but they also showed a certain tendency to produce elevated amounts of IFNγ. FOXP3^+^ Treg cells are a dynamic population that exhibit considerable plasticity [37]. Under a heavily Th1-polarized mucosal environment or highly inflammatory conditions, Treg cells can acquire a Th1-like phenotype that includes the expression of T-bet transcription factor and secretion of IFNγ [38,39]. The acquisition of this Th1-like phenotype is usually associated with a loss of Treg suppressor function [40]. During the T cell transfer model, a very severe inflammatory environment develops in the colon as a result of enteric antigen-driven differentiation and expansion of Th1 and Th17 cells [36]. Such environments could skew some of the newly differentiated *Aim2^−/−^* FOXP3^+^ cells into IFNγ-producing Th1-like Tregs, explaining the slight elevation in IFNγ production. This could also partly explain the fact that mice receiving *Aim2^−/−^* naïve T cells show areas of severe intestinal inflammation, despite having a higher number of FOXP3^+^ cells in the spleen and colon. It seems clear that this increase in the FOXP3^+^ population is still not equivalent to the “Treg cotransfer” situation, and is not enough to fully prevent disease development in this highly colitogenic model.

The molecular mechanism whereby AIM2 exerts its function in CD4^+^ T cells remains to be fully elucidated. AIM2 has been shown to physically inhibit the activation of the phosphatidylinositol 3-kinase (PI3K)/AKT signaling pathway independently of the inflammasome [22,41]. The role of the PI3K/AKT pathway in Treg cell differentiation, stability and function is complex and can vary between tTreg and pTreg cells. Some components of the PI3K/AKT pathway, such as forkhead box protein 1 (FOXO1) and the phosphoinositide-dependent kinase 1 (PDK1), appear necessary for tTreg and pTreg development and function, while others, such as AKT or the mechanistic target of rapamycin complex 1 (mTORC1), seem to block it (reviewed in [42]). Recently, AIM2 has been also shown to support extracellular signal-regulated kinase 1/2 (ERK1/2) phosphorylation and activation independently of the inflammasome in non-small cell lung cancer (NSCLC) cells [43]. Of note, ERK1/2 is an important regulator of CD4^+^ T cell development and differentiation that was shown to suppress FOXP3^+^ Treg cell development [44,45,46,47]. Thus, fully uncovering how AIM2 affects the PI3K/AKT or ERK1/2 pathways in CD4^+^ lymphocytes may provide useful information to understand the intricacy of signals that contribute to Treg cell development and function.

A main limitation of our study is the limited amount of human data. The role of AIM2 in human CD4^+^ T cells is still unknown and further studies should be performed in order to confirm that the results obtained in mouse cells can be extrapolated to human cells. In addition, it would be important to investigate whether the T cell-intrinsic function of AIM2 affects autoimmune diseases and other immune-related diseases, including cancer and the anti-tumor immune response. As an example, it would be interesting to see whether genetic variations (methylation status, single nucleotide polymorphisms, etc.) affecting AIM2 gene expression or function are associated with the susceptibility to Treg-associated diseases in humans, such as inflammatory bowel disease.

Altogether, we show that AIM2 function is not confined to innate immune cells but is also important in adaptive CD4^+^ T cells. Our results identify AIM2 as a potential regulator of Treg cell differentiation and as an intervention target for restoring T cell homeostasis.

## 4. Materials and Methods

### 4.1. Antibodies, Primers and Common Reagents

A complete list of common reagents, antibodies and primers is included as Appendix A.

### 4.2. Mice

Specific pathogen-free (SPF) C57BL/6 mice (wild type; WT) and mice deficient in AIM2 (*Aim2^−/−^*) on the C57BL/6 background were originally purchased from Jackson Laboratories (Stock #013144). Then, all mice were bred under SPF conditions in filter top cages with standard food and water ad libitum at the animal facility of the University Miguel Hernández (UMH). Mice were cohoused after weaning to minimize the effect of potential microbiome differences, and all experiments were performed with sex- and age-matched animals. The Office for Responsible Research of the UMH approved all experimental procedures, assuring that all animals received humane care according to the Spanish and European legislations.

### 4.3. Human Samples

Peripheral blood was obtained from 3 healthy volunteers, two females and 1 male, aged between 25 and 42 years. Written informed consent was obtained from each participant and the protocol was supervised and approved by the Clinical Research Ethics Committee of the Alicante General Hospital (HGUA). Peripheral blood mononuclear cells (PBMCs) were isolated by gradient centrifugation using Biocoll solution (Biochrom, Berlin, Germany). CD4^+^ cells were then isolated from PBMCs by immunomagnetic selection using an EasySep^TM^ Human CD4^+^ T cell isolation kit (StemCell Technologies, Vancouver, BC, Canada).

### 4.4. CD4^+^ T Cell Isolation, Culture, Stimulation and Differentiation 

Complete RPMI 1640 or IMDM medium (Life Technologies, Carlsbad, CA, USA) supplemented with 10% heat-inactivated fetal calf serum, 2 mM L-glutamine, 100 U/mL penicillin and 100 mg/mL streptomycin was used throughout the experiments.

For AIM2 expression experiments, CD3^+^CD4^+^TCRβ^+^ cells were FACS-sorted using a FACSAria III flow cytometer with FACSDiva software (BD Biosciences, San Jose, CA 95131, USA). The purity of the sorted population was higher than 99%.

For CD4^+^ T cell stimulation, cells were isolated from a single-cell suspension of splenocytes or mesenteric lymph node (MLN) cells by immunomagnetic negative selection using the EasySep^TM^ mouse CD4^+^ T cell isolation kit (StemCell Technologies, Vancouver, BC V6A 1B6, Canada). The purity of the enriched population was above 90% in all experiments. After enrichment, cells were cultured in complete RPMI medium and stimulated with 5 μg/mL of plate-bound anti-CD3 and 1 μg/mL of soluble anti-CD28 antibodies. Twenty-four hour culture supernatants were collected for cytokine analysis using Ready-Set-Go ELISA kits (Life Technologies, San Diego, CA 92008, USA).

For T cell differentiation, naïve CD4^+^ T cells (CD44^low^CD62L^high^) were isolated by immunomagnetic selection using EasySep^TM^ mouse CD4^+^ naïve T cell isolation kit (StemCell Technologies, Vancouver, BC, V6A 1B6, Canada). The purity was higher than 93% in all experiments. Cells were then cultured in 48-well plates and stimulated in the presence of Th1, Th17 or Treg polarizing conditions. For Th1 differentiation, recombinant mouse IL-12 (10 ng/mL) and neutralizing anti-IL-4 antibody (10 mg/mL) were added into the culture. After 2 days, recombinant mouse IL-2 (20 U/mL) was added into the Th1 culture. For Th17 differentiation, recombinant mouse IL-6 (20 ng/mL), recombinant human TGFβ (4 ng/mL), anti-IL-4 antibody (10 mg/mL) and anti-IFNγ antibody (10 mg/mL) were added. For Treg cell differentiation, recombinant human TGFβ (10 ng/mL) and recombinant mouse IL-2 (100 U/mL) were added. After 5 days, cells were re-stimulated for intracellular IL-4, IL-17A and IFNγ detection, as detailed below. FOXP3 expression after Treg differentiation was analyzed at days 3 and 5 without re-stimulation.

### 4.5. Treg Suppression Assay

Naïve (CD4^+^CD45RB^high^CD25^−^) T cells and Treg cells (CD4^+^CD45RB^low^CD25^+^) were isolated from a single-cell suspension of splenocytes by immunomagnetic selection and FACS sorting. After sorting, naïve T cells were labeled with the CellTrace Far Red Cell Proliferation Kit (Thermo Fisher Scientific, Waltham, USA) following the manufacturer’s instructions and adjusted to 5 × 10^5^ mL^−1^ in complete RPMI culture media. Unlabeled Treg cells were adjusted to 2.5 × 10^5^ mL^−1^ in complete RPMI culture media. Cells were then co-cultured in a round-bottom 96-well plate at Treg/Tnaïve cell ratios of 1:2, 1:4, 1:8, 1:16 and 1:32. Finally, cells were stimulated using a Dynabeads Mouse T-Activator CD3/CD28 kit (Life Technologies, Carlsbad, CA, USA) following the manufacturer’s instructions in the presence of 10 ng/mL of IL-2. After 72 h, the cells were collected and proliferation of naïve T cells was analyzed according to CellTrace Far Red fluorescence by flow cytometry.

### 4.6. Flow Cytometry Analysis

Flow cytometry data were acquired in a BD FACSCanto^TM^ (BD Biosciences, San Jose, CA 95131, USA) and analyzed with FlowJo software (Tree Star, Ashland, OR 97520, USA). For the measurement of intracellular cytokines, CD4^+^ T cells were stimulated for 5 h with anti-CD3 (5 mg/mL) and anti-CD28 (1 mg/mL) antibodies in the presence of brefeldin A (Life Technologies, San Diego, CA 92008, USA). Surface and intracellular markers were analyzed using antibodies to CD3, CD4, TCRβ, IL-4, IL-17A, IFNγ, AIM2 or FOXP3 (Appendix A) according to the manufacturer’s instructions. Isotype control antibodies and fluorescence minus one (FMO) controls were used as controls in all experiments. All analyses were performed following the guidelines for the use of flow cytometry and cell sorting in immunology [48].

### 4.7. Isolation of RNA and Quantitative Real-Time PCR

RNA isolation was performed using a RNeasy Mini Kit (Qiagen, Hilden, Germany) according to the manufacturer’s instructions. Then, to avoid unwanted DNA amplification, all samples were pre-treated with a DNA-free DNA removal kit (Thermo Fisher Scientific, Waltham, MA, USA). For gene expression analysis, 10 ng of RNA sample was then used for one-step qPCR using a qScript One-Step RT-qPCR kit (Quantabio, Beverly, MA, USA). GAPDH expression was used as an internal reference in all experiments, and all samples were tested in duplicate. The qRT-PCR primers for each specific target gene were designed based on their reported sequence (Appendix A). The calculation of mRNA fold induction was performed using the double delta Ct (cycle threshold) method. The average delta Ct of the control group in each experiment was used as the reference value to calculate the double delta Ct for each individual sample in each group.

### 4.8. Immunoblot

For Western blot analysis, cells were lysed in total cell lysis buffer containing 0.15 M NaCl, 10 mM HEPES, 0.1 mM EDTA, 0.1 mM EGTA, 1 mM NaF, 1 mM Na3VO4, 10 mM KCl, 0.5% NP-40 and protease inhibitor cocktail (10% vol/vol). Proteins (20 μg/lane) were then boiled at 95 °C in the presence of LDS sample buffer and 2-mercaptoethanol, subjected to SDS PAGE and then transferred to Immuno-Blot PVDF membranes (Bio-Rad, Hercules, CA, USA). Membranes were blocked for 30 min in 3% BSA and 0.05% Tween 20 in PBS and incubated overnight with the corresponding primary antibodies (Appendix A). After washing, the membranes were then incubated for 1 h at room temperature with the corresponding anti-mouse or anti-rabbit IgG-HRP secondary antibody. The activity of membrane-bound peroxidase was detected using Supersignal West Femto Chemi and Supersignal West Pico Chemi (Thermo Fisher Scientific, Waltham, MA, USA) and scanned in a ChemiDoc XRS+ (BioRad, Hercules, CA, USA).

### 4.9. Quantification of Cytokine Levels

Enzyme-linked immunosorbent assays (ELISA) for the quantitative measurement of cytokine production were performed using Ready-SET-Go ELISA kits (Thermo Fisher Scientific, San Diego, CA, USA) according to the manufacturer’s instructions. All samples were tested in triplicate. Standard curves were generated for each plate, and the average optical density of the zero standard was subtracted from the rest of the standards and samples to obtain a corrected concentration for all cytokines.

### 4.10. RNAseq Analysis

RNA was extracted from splenic CD4^+^ T cells using a RNeasy Mini Kit (QIAgen, Hilden, Germany) following the manufacturer’s instructions. The quantity and quality of the RNA were evaluated using a Qubit RNA HS Assay Kit (Thermo Fisher Scientific, San Diego, CA, USA) and Agilent RNA 6000 Nano Chips (Agilent Technologies, Santa Clara, CA, USA), respectively. Sequencing libraries were prepared following the TruSeq Stranded mRNA Sample Preparation Guide (Part # 15031047 Rev. E) with the corresponding kit (Illumina Inc. Cat.# RS-122-2101 or RS-122-2102 (set A or B, respectively)), starting from 125 ng of total RNA per sample. Sequencing of the mRNA libraries was performed in a HiSeq2500 (Illumina Inc, San Diego, CA, USA) to obtain at least 30 million 51nt-single-end reads per sample. FASTQ files’ reads were aligned to the genome dat base (Mus_musculus.GRCm38.94.gtf) by STAR software [49], then read counts per gene were calculated. Genes with no counts in any sample were excluded for the analysis, as they were considered to be non-expressed in the studied system. The STAR program against Mus musculus genome (GRCm38) was used to map the reads followed by the quantification of genes with the RSEM software [50] using GENCODE m15 reference annotation [51]. After removing genes with an expected value greater than ten, we used the TMM method and limma-voom transformation [52] to normalize the non-biological variability. Differential expression between different groups was assessed using moderated t-statistics. Gene Ontology and Reactome canonical pathway enrichment analyses were performed through the GSEA function in the clusterProfiler package (gseGO and gsepathway) using previously computed t-statistic values. Additionally, we evaluated certain selected gene sets (REF). Heatmaps and principal component plots were performed using R statistical software.

### 4.11. Induction of Colitis by Adoptive Transfer of Naïve T Cells

Splenocytes from WT and *Aim2^−/−^* donor mice were enriched for CD4^+^ T cells by negative immunomagnetic isolation, as described above. Cells were then stained with anti-CD4, anti-CD25 and anti-CD45RB antibodies (Appendix A) and sorted into naïve CD4^+^CD45RB^high^CD25^−^ and regulatory CD4^+^CD45RB^low^CD25^+^ populations (purity > 99%) with a FACSAria III flow cytometer using FACSDiva software (BD Biosciences, San Jose, CA, USA). Seven- to ten-week-old RAG1-deficient mice (*Rag1^−/−^*) were used as recipients. Sex-matched mice were reconstituted with 5 × 10^5^ FACS-sorted CD45RB^high^CD25^−^ naïve T cells from WT or *Aim2^−/−^* mice by intraperitoneal injection. In the cotransfer groups, mice were co-injected with the naïve CD4^+^ T cell population from WT mice plus 2 × 10^5^ CD45RB^low^CD25^+^ Tregs from either WT or *Aim2^−/−^* mice. After reconstitution, mice were monitored for signs of intestinal inflammation such as weight loss and diarrhea. Diseased animals were sacrificed for analysis 15 weeks after transfer.

### 4.12. Histology, Immunohistochemistry and Evaluation of Colitis

The entire colon was excised, opened longitudinally, rolled onto a wooden stick, fixed in 10% buffered formalin and embedded in paraffin wax. Five-micrometer tissue sections were obtained for hematoxylin and eosin (H&E) staining or immunohistochemistry. The histopathological score was calculated as described [36], based on the degree of inflammation in the lamina propria (score 0–4), Globet cell loss (score 0–2), abnormalities in the crypt structure (score 0–2), presence of crypt abscesses (score 0–2) and transmural infiltration (score 0–2). Histological evaluations were performed in a blinded fashion by a certified pathologist (G.P.). The detection of FOXP3 by immunohistochemistry was carried out via deparaffinization and rehydration of the tissue section with xylene and decreasing strengths of alcohols (100%, 96% and 70% ethanol) followed by water. Antigen retrieval was performed by heating in a microwave with a solution of sodium citrate 1 M, pH 6. Blocking of the endogenous peroxidase, the non-specific binding of the antibody and the non-specific endogenous signal from avidin and biotin was performed following standard protocols. For the incubation with the primary antibody, a dilution of 1:100 of anti-FOXP3 monoclonal antibody overnight (FJK-16s, eBioscience, San Diego, CA, USA) was performed, followed by incubation with a 1:200 dilution of the secondary antibody (biotinylated goat anti-Rat IgG from Vector Laboratories, BA-9400). To amplify the signal, the peroxidase Avidin–Biotin Complex kit was used (Vectastain Peroxidase Standard ABC kit, Vector Laboratories) and the signal was detected with diaminobenzidine (DAB) peroxidase substrate kit (Vector Laboratories, Burlingame, CA, USA), followed by dehydration of the tissue and mounting with Eukitt (Panreac, Barcelona, Spain).

### 4.13. Statistical Analysis

The normality of the distribution of quantitative data was determined by D’Agostino–Pearson and Shapiro–Wilk normality tests whenever applicable. In cases of loss of normality, statistical differences between groups were analyzed using the nonparametric Mann–Whitney U test or the Kruskal–Wallis test with Dunn’s multiple comparisons test, and the data are displayed as medians ± interquartile ranges. Variables with normal distribution of sample data were analyzed using unpaired t-test with Welch’s correction or one-way ANOVA with Holm–Sidak multiple comparison test, and the data are displayed as means ± standard deviations. All *p* values were two-tailed and *p* values lower than 0.05 were considered significant. All calculations were performed using GraphPad Prism 6.0.

## Figures and Tables

**Figure 1 ijms-23-02230-f001:**
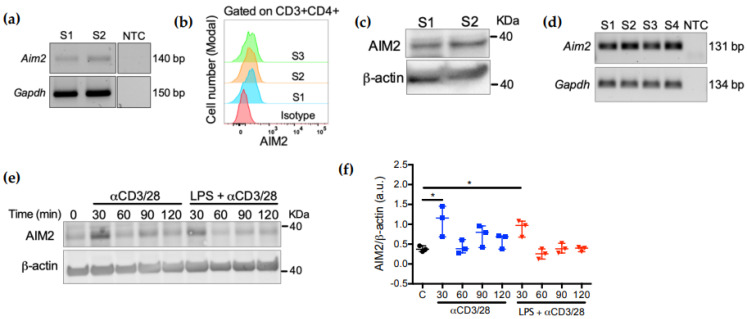
AIM2 expression is controlled by the TCR. (**a**) Total RNA samples from two different subjects (S1 and S2) were subjected to qRT-PCR for the amplification of *Aim2* mRNA and visualized on an agarose gel after 30 cycles. NTCs were run on the same gel but were not contiguous. (**b**) Flow cytometry analysis of AIM2 expression in CD3^+^CD4^+^ T cells from peripheral blood mononuclear cells (PBMCs) of three different subjects (S1–S3). (**c**) Western blot analysis of AIM2 expression in CD4^+^ cells purified from PBMCs of two different subjects (S1–S2). (**d**) Total RNA samples from four different mouse CD4^+^ T cell samples (S1–S4) were subjected to qRT-PCR for the amplification of *Aim2* mRNA and visualized on an agarose gel after 30 cycles. (**e**) Western blot analysis of AIM2 expression in mouse spleen CD4^+^ T cells. CD4^+^ T cells were stimulated with anti-CD3 and anti-CD28 (αCD3/28) antibodies with or without LPS pre-stimulation (2 h) and total protein extracts were collected at the indicated timepoints. Anti-mouse AIM2 and anti-mouse β-actin primary antibodies were used after immunoblot for detection of AIM2 expression and loading control, respectively. (**f**) Quantification of blot images performed using ImageJ software (au; arbitrary units). Data were analyzed using Sidak’s multiple comparison test and are displayed as scatter plot with median ± range; * *p* < 0.05. Data are representative of three different experiments with similar results. NTC = non-template control.

**Figure 2 ijms-23-02230-f002:**
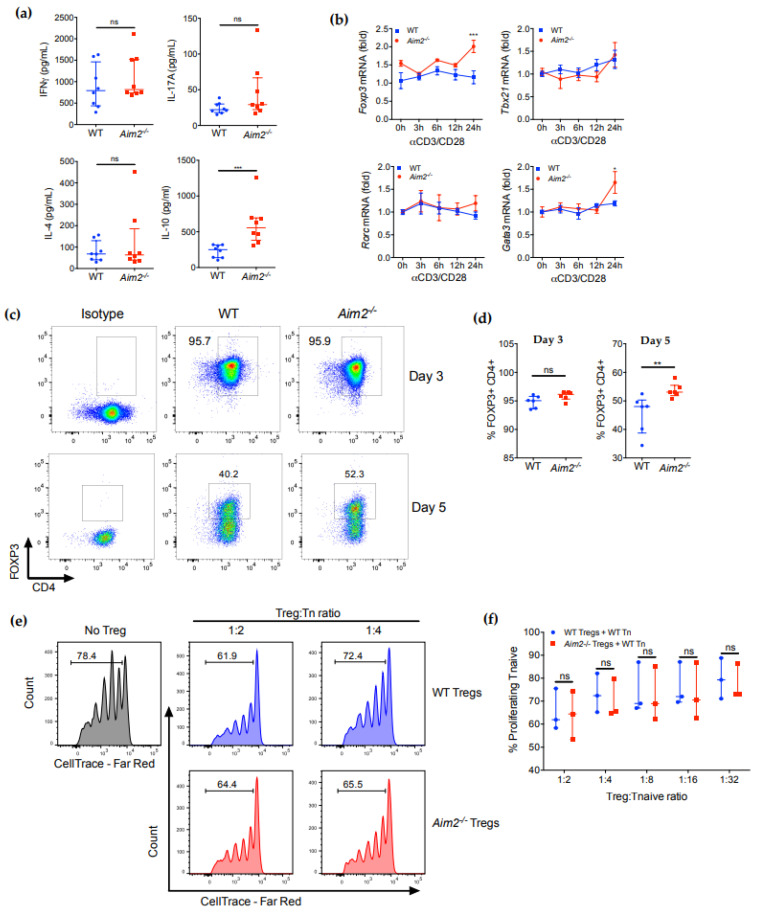
Lack of AIM2 promotes FOXP3 expression in vitro but does not affect the suppressive ability of Treg cells. (**a**) Cytokine levels in culture supernatants of splenic CD4^+^ T cells from WT or *Aim2^−/−^* mice stimulated with anti-CD3/CD28 antibodies for 24 h. (**b**) RT-qPCR analysis of *Foxp3*, *Tbx21*, *Gata3* and *Rorc* mRNA expression in splenic CD4^+^ T cells isolated from WT or *Aim2^−/−^* mice (*n* = 4/group) and stimulated with anti-CD3/CD28 (αCD3/28) antibodies for the indicated timepoints. (**c**) Flow cytometry analysis of FOXP3 expression in naïve CD4^+^ T cells from WT or *Aim2^−/−^* mice cultured under Treg polarizing conditions for 3 or 5 days. Data are representative of two experiments with a total of *n* = 6/group. (**d**) Quantification of FOXP3^+^ cells from the flow cytometry analysis shown in (**c**). (**e**) Flow cytometry of the proliferation of naïve T cells (Tn) in the absence of Treg cells (No Treg) or in the presence of Treg cells at different ratios. (**f**) Quantification of proliferating T cells from the flow cytometry analysis shown in (**e**). Note: * *p* < 0.05, *** p* < 0.01, **** p* < 0.001. The data were analyzed using the Mann-Whitney U test (**a**,**d**) or two-way ANOVA with Sidak’s multiple comparison test (**b**,**f**) and are displayed as mean ± standard deviation (**b**) or as median ± interquartile range (**a**,**d**,**f**).

**Figure 3 ijms-23-02230-f003:**
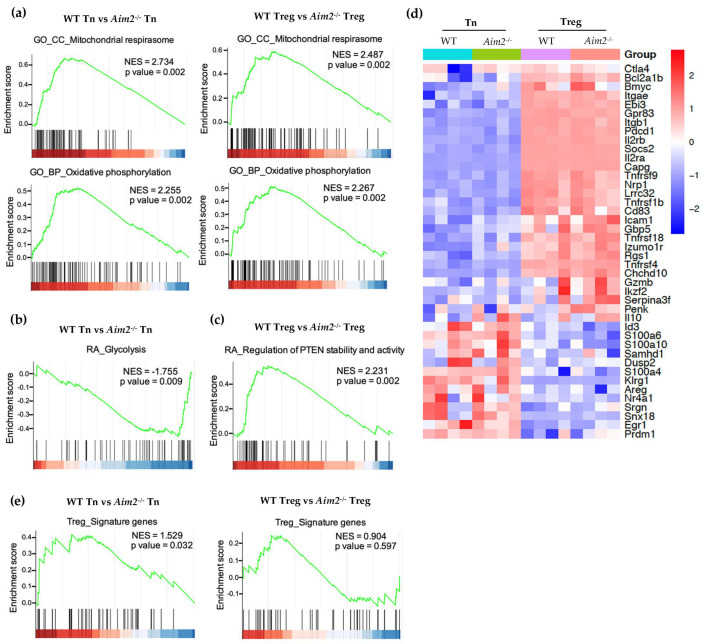
RNAseq analysis in naïve T cells and during Treg polarization conditions. (**a**–**c**) Gene set enrichment analysis (GSEA) of CD4^+^ naïve T cells (Tn) from WT and *Aim2^−/−^* and from WT and *Aim2^−/−^* T cells stimulated under Treg polarizing conditions (Treg) for 2 days (*n* = 4/group). GO = gene ontology database; CC = cellular component; BP = biological process; NES = normalized enrichment score; RA = reactome database. (**d**) Heatmap showing the expression of Treg-associated genes in Tn and Treg from WT and *Aim2^−/−^* mice (*n* = 4/group). Color coded values represent log2 values of the normalized counts, mean-centered for each gene. (**e**) GSEA analysis of Treg-associated genes in Tn (left) and Treg (right) cells from WT and *Aim2^−/−^* mice (*n* = 4/group).

**Figure 4 ijms-23-02230-f004:**
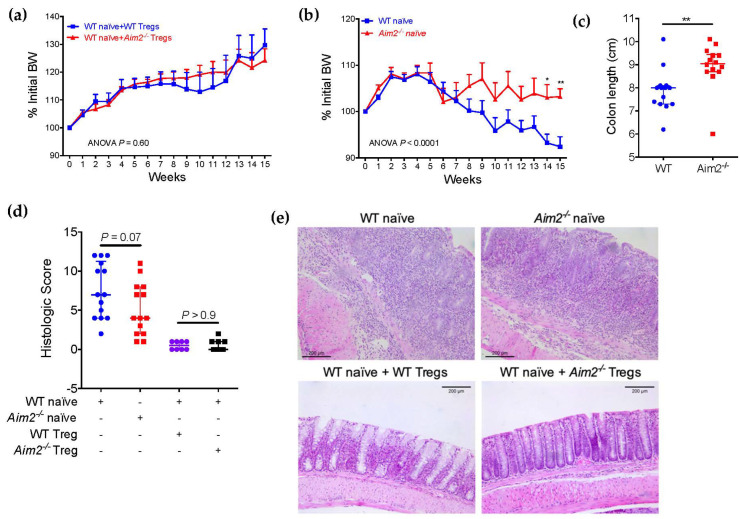
*Aim2^−/−^* naïve T cells induce less body weight loss in the adoptive transfer model. *Rag1^−/−^* recipient mice were transferred with FACS-sorted naïve CD4^+^ T cells (CD4^+^CD45RB^high^CD25^−^) from WT (*n* = 14) or *Aim2^−/−^* (*n* = 14) mice. Cotransfer groups received WT naïve T cells plus CD4^+^CD45RB^low^CD25^+^ regulatory T (Treg) cells from either WT (*n* = 8) or *Aim2^−/−^* (*n* = 10) mice. Mice were then monitored for colitis induction and sacrificed 15 weeks after transfer. (**a**,**b**) Percentages of initial body weight (BW) in the cotransfer groups (**a**) or in the *Rag1^−/−^* mice transferred only with naïve T cells from WT or *Aim2^−/−^* donors (**b**). (**c**) Measurement of colon length in mice transferred with naïve T cells from WT or *Aim2^−/−^* donors. (**d**) Quantification of the histopathologic score of colitis in each group of mice. (**e**) Hematoxylin and eosin staining of colons from each group of recipient mice (original magnification, ×10). Scale bars represent 200 μm. (**a**,**b**) Data are from one experiment representative of two independent experiments. (**c**,**d**) Pooled data from two independent experiments. Note: * *p* < 0.05, ** *p* < 0.01. Data were analyzed using a two-way ANOVA with Holm–Sidak multiple comparison test and are displayed as mean ± standard deviation (**a**,**b**), or using a Mann-Whitney U test and displayed as median ± interquartile range (**c**,**d**).

**Figure 5 ijms-23-02230-f005:**
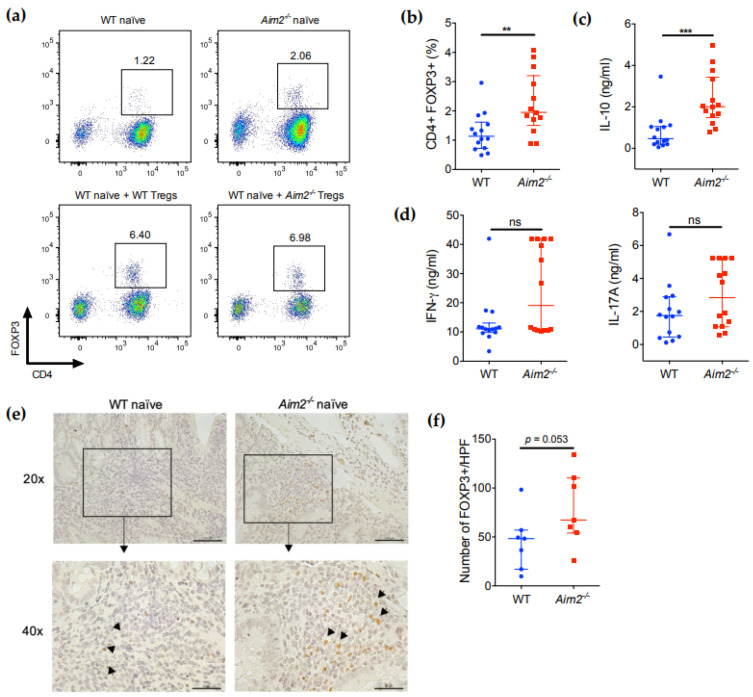
*Aim2^−/−^* naïve T cells show increased differentiation into FOXP3^+^ cells in vivo. (**a**) Flow cytometry analysis of FOXP3 expression in splenic CD4^+^ cells isolated from *Rag1^−/−^* mice 15 weeks after transfer of naïve T cells (CD4^+^CD45RB^high^CD25^−^) from WT (*n* = 14) or *Aim2^−/−^* (*n* = 14) mice, or co-transferred with WT naïve T cells plus CD4^+^CD45RB^low^CD25^+^ regulatory T (Treg) cells from either WT (*n* = 8) or *Aim2^−/−^* (*n* = 10) mice. Images are representative of two independent experiments. (**b**) Quantitative analysis of the percentages of FOXP3^+^ cells in the spleens of *Rag1^−/−^* mice after transfer of naïve T cells from WT or *Aim2^−/−^* mice. (**c**,**d**) Cytokine levels in culture supernatants of CD4^+^ T cells isolated from the spleens of *Rag1^−/−^* mice after transfer of WT or *Aim2^−/−^* naïve T cells as described in (**a**). Cells were stimulated with anti-CD3/CD28 antibodies for 24 h. (**e**) Immunohistochemistry analysis of FOXP3^+^ cells in colonic tissue of *Rag1^−/−^* mice receiving either WT or *Aim2^−/−^* naïve T cells as described in (**a**). Scale bars represent 100 μm (top photographs, original magnification ×20) or 50 μm (bottom photographs, original magnification ×40). Data shown are representative of one experiment with 7 mice per group. (**f**) Quantitative analysis of the immunohistochemistry analysis described in (**e**). Note: ** *p* < 0.01, *** *p* < 0.001, ns = not significant. Data were analyzed using a Mann-Whitney U test and displayed as median ± interquartile range (**b**–**d**,**f**).

## Data Availability

The datasets generated during the RNA sequencing (RNAseq) analysis are available at the GEO repository ID: GSE169615. The rest of the datasets used or analyzed during the current study are available from the corresponding author on reasonable request.

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
