# Peer review of "Absent in Melanoma 2 (AIM2) Regulates the Stability of Regulatory T Cells"

_ijms, 2022, doi:10.3390/ijms23042230_

Round 1

Reviewer 1 Report

In this manuscript, the authors analyzed AIM2 impact in CD4+ T cells, with a special focus on Treg and a mouse model of colitis. It is noteworthy that another manuscript published recently analysed similar parameters (Chou et al., Nature 2021), and yielded opposite results. Consequently, this manuscript is confusing, but it could be of interest if authors improve some parts of the manuscript. Below are specific comments:
To show that AIM2 is controlled by TCR activation, authors should show quantification data from western blots that includes biological replicates.
Furthermore, it would be informative to know whether AIM2 is expressed identically between the different in vitro generated CD4+ T cell subtypes (Th1, Th2, Th17, Treg): Western blot with quantifications + qPCR (showing fold change from Ct calculations). Even if Chou et al already showed some data for Th1 and Th17, how AIM2 impacts Th1/Th2/Th17 would also be of intersest (especially given that they show opposite results while there is no data for Th2 - moreover Figure 2C shows an increase in GATA3 at 24h).
Data showing total NFAT protein levels by western blot is not a good surrogate to assess T-cell activation status, and thus to decipher whether AIM2 is (not) impeding T-cell activation. If one is willing to use NFAT to this purpose, phosphorylation status or nuclear translocation of the protein should instead be assessed.
In line whith this comment, it would be good to assess activation threshold of T cells AIM2 KO vs WT using a dose-response of anti-CD3 (coated + 1 fixed dose of soluble anti-CD28). This would help to decipher whether suboptimal activation (which is physiologically more relevant) would lead to more or less activation (readout could be cytokine secretion and/or proliferation). Indeed, a supraoptimal stimulus could prevent one to notice differences.
The improved stability of differentiated Treg is not striking. As an exemple the y-axis of Figure 2E, 5 days, has been cropped to try making readers appreciate the difference. One question that arises is whether the difference would be more obvious at later time points (day 7 and/or 9). Also, could it be possible that the (slight) difference authors observed is due to a decrease in cell expansion? What is the expansion rate of AIM2 KO vs WT upon 5 days using the iTreg differentiation protocol ?
It would be of interest to see whether induced Treg in AIM2 KO vs WT have the same suppressive activity in vitro. A dose response (ratio Treg:Teff) would be good in order to see whether suboptimal number of iTreg give the same results between WT and AIM2 KO.
Increase of Treg in spleen in the in vivo model of colitis is not convincing since the FoxP3- cells in the AIM2 KO cell transfer group look to have higher fluorescence background in FoxP3 channel. This leads to a FoxP3+ gating that includes what looks to be FoxP3- cells (few, but that are artificially increasing FoxP3+ fraction). Apart from increased Treg differentiation in vivo (not supported in vitro) or increased stability of Treg in vivo, it is possible that differentiation into pathologic subtypes (Th1/Th17) is lowered and/or that Th2 differentiation is increased. As an exemple, authors may try to analyze mRNA from spleen and look at key transcription factors and cytokine associated with the various Th subtypes.

The fact that Chou et al. showed the exact opposite of what authors claim will be confusing to readers. It should be developped a bit more.

It is not entirely clear why authors moved from human T cells to mouse T cells in the first part.
I believe there is a mistake in material and method section: doses of anti-CD3 and anti-CD28 for T-cell activation look abnormally high (mg/ml -> ug/mL).

Author Response

Reviewer 1
In this manuscript, the authors analyzed AIM2 impact in CD4+ T cells, with a special focus 
on Treg and a mouse model of colitis. It is noteworthy that another manuscript published 
recently analysed similar parameters (Chou et al., Nature 2021), and yielded opposite results. 
Consequently, this manuscript is confusing, but it could be of interest if authors improve 
some parts of the manuscript. Below are specific comments:
To show that AIM2 is controlled by TCR activation, authors should show quantification data 
from western blots that includes biological replicates.

We have added the quantification of three different western blots in Figure 1f.
Furthermore, it would be informative to know whether AIM2 is expressed identically 
between the different in vitro generated CD4+ T cell subtypes (Th1, Th2, Th17, Treg): 
Western blot with quantifications + qPCR (showing fold change from Ct calculations). 
We followed the reviewer suggestion and analyzed the expression of AIM2 in Treg cells and
different Th subsets. We analyzed AIM2 protein by flow cytometry. Intriguingly, a similar 
percentage of AIM2-positive cells was detected in Treg, Th1, Th2 and Th17 cells. These 
results also diverge from the report by Chou et al (Nature 2021), in which they show a higher 
mRNA expression in Treg cells when compared with conventional T cells (bulk CD4+). This 
difference could be due to the fact that we are measuring protein instead of mRNA and/or
due to the different origin of the Tregs (in vitro generated vs isolated from mice in the report 
by Chou et al.).

The data is now displayed as Figure S5 and explained in the results section 2.2 (lines 141-
145) of the revised manuscript. We also discuss the disparity with the paper by Chou et al. in 
the Discussion section (lines 277-280).
Even if Chou et al already showed some data for Th1 and Th17, how AIM2 impacts 
Th1/Th2/Th17 would also be of intersest (especially given that they show opposite results 
while there is no data for Th2 - moreover Figure 2C shows an increase in GATA3 at 24h).
In the original submission we also presented data showing that AIM2 does not affect Th1 and 
Th17 differentiation (previous supplementary Figure 3). We have repeated the experiments 
including the analysis of Th2 differentiation following the reviewer’s suggestion. As in Th1 
Response to reviewers – Manuscript ID: ijms-1405637

and Th17 conditions, we did not find differences on Th2 differentiation between WT and 
Aim2-/- naïve T cells. These data are described in the Results section (see lines 132-135) and
displayed as Figure S4a-c on the revised manuscript.
Data showing total NFAT protein levels by western blot is not a good surrogate to assess Tcell activation status, and thus to decipher whether AIM2 is (not) impeding T-cell activation. 
If one is willing to use NFAT to this purpose, phosphorylation status or nuclear translocation 
of the protein should instead be assessed.

We used NFATc1 because it is an inducible member of the NFAT family, whose expression 
is controlled by TCR stimulation. Nevertheless, since the cytokine data shown in Figure 2 
and the new data assessing the activation threshold of Aim2-/- T cells (Figure S3 – see 
response to the next comment below) clearly show that functional TCR activation is 
unaffected by the absence of AIM2, we reason that the western blot data provides somewhat 
redundant information and we have decided to remove it from the manuscript.
In line whith this comment, it would be good to assess activation threshold of T cells AIM2 
KO vs WT using a dose-response of anti-CD3 (coated + 1 fixed dose of soluble anti-CD28). 
This would help to decipher whether suboptimal activation (which is physiologically more 
relevant) would lead to more or less activation (readout could be cytokine secretion and/or 
proliferation). Indeed, a supraoptimal stimulus could prevent one to notice differences.
We followed the reviewer’s suggestion and stimulated WT and Aim2-/- T cells with different 
concentrations of anti-CD3 (0.5 µg/mL, 1 µg/mL, 2 µg/mL and 5 µg/mL) in the presence of a 
fixed dose of anti-CD28. These data are shown in supplementary Figure S3 and the 
description of the results is included in the revised version of the manuscript (lines 121-123). 
Briefly, the data shows that even at low concentrations of anti-CD3, AIM2-deficient T cells 
show normal production of IL-2 or IFNg, but still enhanced production of IL-10.
The improved stability of differentiated Treg is not striking. As an exemple the y-axis of 
Figure 2E, 5 days, has been cropped to try making readers appreciate the difference. One 
question that arises is whether the difference would be more obvious at later time points (day 
7 and/or 9).

Also, could it be possible that the (slight) difference authors observed is due to a 
decrease in cell expansion? What is the expansion rate of AIM2 KO vs WT upon 5 days 
using the iTreg differentiation protocol?
The range and interval of the y-axis was determined automatically by the statistical software 
(GraphPad Prism). Nevertheless, we agree on the fact that the improved stability is not 
striking (around 20%), but it is significant and very consistent intra- and inter-experiment.
We appreciate the suggestion of analyzing these data at later timepoints. In fact, we did check 
Foxp3 expression at day 7 before the original submission of the manuscript, but cell viability 
dropped drastically at this timepoint and therefore we did not use the data.

Last, data shows percentage of Foxp3+ cells over total CD4+ cells, thus it should not be 
affected by cell expansion. Nevertheless, based on cell counting on Neubauer chamber, the 
expansion rate seemed very similar between WT and Aim2-/- cells (usually 2-3 fold increase 
from the initial number of cells in each well).

It would be of interest to see whether induced Treg in AIM2 KO vs WT have the same 
suppressive activity in vitro. A dose response (ratio Treg:Teff) would be good in order to see 
whether suboptimal number of iTreg give the same results between WT and AIM2 KO.
We thank the reviewer for this comment. Indeed, this is an important experiment so we have 
followed this suggestion and performed an in vitro suppression assay to test the ability of 
AIM2-deficient Treg cells to inhibit the proliferation of naïve T cells. The data is shown in 
Figure 2e-f in the revised manuscript and the results are introduced in the text (lines 138–
141). Our data shows that WT and Aim2-/- Treg cells are equally suppressive at all ratios 
tested (1:2 to 1:32).

Increase of Treg in spleen in the in vivo model of colitis is not convincing since the FoxP3-
cells in the AIM2 KO cell transfer group look to have higher fluorescence background in 
FoxP3 channel. This leads to a FoxP3+ gating that includes what looks to be FoxP3- cells 
(few, but that are artificially increasing FoxP3+ fraction). 

We respectfully disagree with this comment. We believe that the differences between groups, 
albeit not dramatic (median is around 2-fold higher in Aim2-/-), are consistent and significant 
(as shown in Figure 5b). Nevertheless, to improve the clarity of the data, we have further 
adjusted all the gates on the Aim2-/- naïve transfer group for that particular experiment. The 
differences are still significant, as shown in Figure 5a and 5b of the revised manuscript. 
Apart from increased Treg differentiation in vivo (not supported in vitro) or increased 
stability of Treg in vivo, it is possible that differentiation into pathologic subtypes 
(Th1/Th17) is lowered and/or that Th2 differentiation is increased. As an exemple, authors 
may try to analyze mRNA from spleen and look at key transcription factors and cytokine 
associated with the various Th subtypes.

In Figure 2c (Figure 2b in the revised manuscript) we did show mRNA expression of Foxp3, 
Gata3, Tbx21 and Rorc genes in CD4+ cells from spleen of WT and Aim2-/- mice at baseline 
and after CD3/CD28 stimulation for different timepoints. Also, in Figure 2b (Figure 2a in the 
revised version) we showed production of classical Th1/Th2/Th17/Treg cytokines from 
splenic CD4+ cells. These data pointed mainly towards differences in Treg cells, not in proinflammatory Th1/Th17 or Th2 subtypes.

We could not perform mRNA analysis of CD4+ cells collected from the spleen of the 
adoptive transfer mice (Rag1-/- mice transferred with WT or Aim2-/- T cells) due to lack of 
sample. Rag1-/- mice have no mature T cells, and even after adoptive transfer the amount of 
CD4+ T cells recovered from spleen is low. We used all the cells for Foxp3 analysis by 
FACS and for in vitro stimulation with CD3/CD28 antibodies to analyze cytokine production. 
Of note, the cytokine production data here again does not seem to support that the differences 
are due to reduced differentiation into pathogenic subtypes, since similar production of 
IFNgamma or IL17 were found between groups (Figure 5d) (in fact, there is a trend towards 
increased IFNg and IL-17 production in CD4+ T cells isolated from the spleen of Rag mice 
that received Aim2-/- T cells).

The fact that Chou et al. showed the exact opposite of what authors claim will be confusing 
to readers. It should be developped a bit more.
The work by Chou et al (Nature 2021, 591, 300–305) is a very thorough study that was 
published while our manuscript was under review at a different journal. We agree with the 
reviewer and we admit that it is also confusing to us. 

We acknowledge in the Discussion that we do not have a clear explanation for this 
discrepancy, apart from the fact that the mice are genetically different. However, following 
the reviewer’s suggestion, we now discuss some methodological differences between the two 
studies that could partly explain the contrasting results (see lines 274-280 on the revised 
version). First, we used naïve T cells for the in vitro differentiation studies and for the 
RNAseq studies, while Chou et al. used bulk CD4+ cells which could be influenced by 
environmental factors such as the gut microbiota. Second, Chou et al. also use the adoptive 
transfer model of colitis to test the suppressive ability of Aim2-/- Treg cells in vivo, but they 
do not explore whether naïve Aim2-/- T cells induce more or less colitis in this model, or 
whether they differentiate more or less towards Treg cells in vivo within the Rag mice. Last, 
Chou et al. assed AIM2 mRNA expression in natural Treg cells while we analyzed AIM2 
protein expression by flow cytometry in in vitro generated Tregs.

However, as commented above, at this time we do not have a clear answer for the divergent 
results.
It is not entirely clear why authors moved from human T cells to mouse T cells in the first 
part.
The use of primary human CD4+ T cells was intended only to determine whether AIM2 was 
expressed in human T cells, as a way to assess whether the study of AIM2 in mouse CD4+ 
cells could be potentially relevant for human biology. It was not feasible for us to perform 
many more experiments with human T cells, and therefore we moved to mouse T cells for the 
remaining of the work. 

I believe there is a mistake in material and method section: doses of anti-CD3 and anti-CD28 
for T-cell activation look abnormally high (mg/ml -> ug/mL).
The reviewer is right, there is a mistake in the units. The right dose is 5 ug/mL of anti-CD3 
and 1ug/mL of anti-CD28. The mistake has been corrected.
Final comment to Reviewer 1: We thank the reviewer for his/her constructive criticism. The 
manuscript has been improved by the comments and experiments suggested by the reviewer. 

Reviewer 2 Report

In the manuscript entitled, "Absent in melanoma 2 (AIM2) regulates the stability of regulatory T cells," Lozano-Ruiz et al have discovered a role for Aim2 in Treg cells. They start by showing that AIM2 is expressed in CD+ T cells. The go on to show that loss of Aim2 in mice leads to increased Treg differentiation. They also show that expression of several cytokines is not affected by Aim2 loss. They performed RNA-Seq analysis of naive T cells and show that genes involved in metabolism are differentially expressed. Adoptive transfer experiment showed that mice receiving Aim2 to deficient naive T cells lost less weight than those receiving WT naive T cells. Overall, the manuscript is well written, experiments are well controlled and conclusions are supported by data presented in the manuscript.  

Author Response

We thank the reviewer for his/her positive assessment of our manuscript

Reviewer 3 Report

With interest, I read the manuscript ijms-1405637. A piece of nice work, reach in data.

Comments (no special order):

1. Overall, the content of human data is relatively limited and restircted to initial experiments only. Please, address in the Discussion as a limitation and suggest further translational steps for future studies.

2. The Discussion is anyway a bit too ascetic. Please, expand a bit, e.g. by addressing point 1 above, by discussion some potential consequences for human pathophysiology, etc.

3. Lines 68-71. Please, mention the role of epigenetic mechanisms in differentiation of T cells in general not only Tregs (PMID: 28322581 and 22824525).

4. Study subjects. Gender?

5. Figure 1. Human mRNA analysis?

6. Statistical analysis. Some data distribution other than normal and were, therefore, analyzed using non-parametric tests. Whay are they presented as mean +/-SD then. E.g. in Figure 2?

7. A graphical abstract would be welcome.

Author Response

Reviewer 3
With interest, I read the manuscript ijms-1405637. A piece of nice work, rich in data.
Comments (no special order):
1. Overall, the content of human data is relatively limited and restircted to initial experiments 
only. Please, address in the Discussion as a limitation and suggest further translational steps 
for future studies.
We agree with the reviewer. The human data is very limited and further studies should be 
performed in order to extrapolate the results to human biology. This limitation is discussed in 
the revised version of the manuscript (lines 306-312). 
2. The Discussion is anyway a bit too ascetic. Please, expand a bit, e.g. by addressing point 1 
above, by discussion some potential consequences for human pathophysiology, etc.
We thank the reviewer for this suggestion. We have expanded the discussion by pointing out 
the interest in investigating the potential T cell-intrinsic function of AIM2 in human disease
and suggesting some future studies, in particular in inflammatory bowel disease. We also 
comment on some methodological differences between our study and the paper by Chou et al 
(Nature 2021, 591, 300–305) (see lines 274-280 on the revised version of the manuscript). 
3. Lines 68-71. Please, mention the role of epigenetic mechanisms in differentiation of T cells 
in general not only Tregs (PMID: 28322581 and 22824525).
We have added a sentence to state that epigenetic modifications are not a specific trait of 
Treg cells, since they influence the differentiation and function of all CD4 T cell subsets 
(lines 65-67 of the revised manuscript).
4. Study subjects. Gender?
Response to reviewers – Manuscript ID: ijms-1405637
Page 2 of 2
Human subjects were 2 females and 1 male. This information has been added to Materials 
and Methods section 4.3. 
In vitro experiments with mouse T cells were performed with T cells isolated from both male 
and female mice, but they were always sex-matched in each experiment.
5. Figure 1. Human mRNA analysis?
Yes, we did check Aim2 mRNA expression in two out of three of the human subjects (due to 
sample availability). The data is now displayed in Figure 1 (see Figure 1a and lines 85-86 of 
the revised manuscript).
6. Statistical analysis. Some data distribution other than normal and were, therefore, analyzed 
using non-parametric tests. Whay are they presented as mean +/-SD then. E.g. in Figure 2?
The reviewer is right, mean +/- SD is not the best way to present non-parametric data. We 
have changed the graphs to present non-parametric data as median +/- interquartile range. 
This information has been added to the Material and Methods section and to the figure 
legends in Figure 2, 4 and 5. 
7. A graphical abstract would be welcome.
We thank the reviewer for this suggestion. We have created a graphical abstract that we 
submit as supplementary figure 7 (Figure S7). We still do not know the mechanistic details 
that could explain how AIM2 reduces Foxp3 stability. Therefore, the graphical abstract is 
quite simple and limited to our main observations; AIM2 expression is affected by the TCR 
and suppresses the expression of genes associated with oxidative phosphorylation and 
mitochondrial respiration, thus reducing Treg stability.
Final comment to Reviewer 3: We thank the reviewer for his/her positive assessment of our 
manuscript and constructive suggestions. The manuscript has been improved by the 
suggestions made by the reviewer

Round 2

Reviewer 1 Report

The authors made significant improvements.